# Training-Free Dataset Pruning for Polyp Segmentation via Community Detection in Similarity Networks

**Md Mostafijur Rahman**                                    MOSTAFIJUR.RAHMAN@UTEXAS.EDU
**Radu Marculescu**                                                        RADUM@UTEXAS.EDU
*The University of Texas at Austin*

**Editors:** Accepted for publication at MIDL 2025

## Abstract

Recent advances in deep learning have been driven by the availability of larger datasets and more complex models; however, this progress comes at the expense of substantial computational and annotation costs. To address these issues, we introduce a new, training-free dataset pruning method, *PRIME*, targeting polyp segmentation in medical imaging. To this end, *PRIME* constructs a similarity network among images in the target dataset and then applies community detection to retain a much smaller, yet representative subset of images from the original dataset. Unlike existing methods that require model training for dataset pruning, our *PRIME* completely avoids model training, thus significantly reducing computational demands. The reduction in the training dataset reduces 56.2% data annotation costs and enables 2.3× faster training of polyp segmentation models compared to training on the entire annotated dataset, with only a 0.5% drop in the DICE score. Consequently, our *PRIME* enables efficient training, fine-tuning, and domain adaptation across medical centers, thus offering a cost-effective solution for deep learning in polyp segmentation. Our implementation is available at https://github.com/SLDGroup/PRIME.

**Keywords:** Training-free, Dataset pruning, Polyp segmentation, Community detection

## 1. Introduction

Polyp segmentation plays a pivotal role in the early detection and prevention of colorectal cancer, one of the leading causes of cancer-related mortality worldwide. Indeed, accurate segmentation enables clinicians to precisely locate and characterize polyps in colonoscopy images, thus guiding therapeutic decisions and follow-up strategies. Recent advances in deep learning (Ronneberger et al., 2015; Zhou et al., 2018; Fan et al., 2020; Dong et al., 2021; Rahman and Marculescu, 2023a; Wang et al., 2022a; Rahman et al., 2024), have brought remarkable improvements in segmentation performance, but these gains come at the cost of increasingly larger datasets with pixel-level annotations. Such annotations demand substantial time and expertise from medical professionals, thus making data curation expensive and labor-intensive. In addition, the heterogeneous appearance of polyps, varying in size, shape, texture, and contrast, exacerbates the challenge of gathering representative training images to ensure robust model generalization.

While efforts in image segmentation research have focused primarily on improving model performance, less attention has been paid to reducing the annotation burden through efficient data selection. Some earlier efforts address the selection of informative data samples under various learning paradigms, particularly in active learning (Settles, 1995). In these methods, scoring functions, such as Shannon's entropy (Shannon, 1948), variation ratio

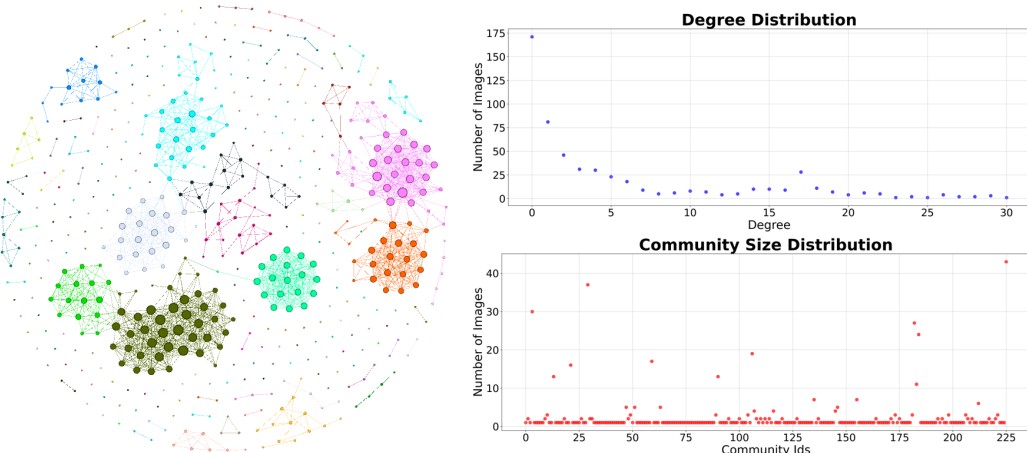

Figure 1: Communities identified with Structural Similarity Index (SSIM) threshold of 0.92 in the CVC-ClinicDB dataset (Bernal et al., 2015). Gephi's algorithm (Blondel et al., 2008) identifies 226 communities with a modularity of 0.843. The degree distribution shows the number of nodes with varying connectivity (0 to 30, left-skewed), while the community distribution highlights the number of nodes per community (0 to <50). By considering top ⌈10%⌉ nodes from each community, we select only 245 (44.5%) representative nodes and prune the rest of the nodes (55.5%). More results are shown in Appendix B.

(Linton, 1965), and Monte Carlo (MC) dropout (Gal and Ghahramani, 2016), are used to identify unlabeled images that maximize the informativeness of the annotated dataset under a limited annotation budget. While these approaches have shown promise in classification tasks (Gal et al., 2017), less effort has been put into semantic segmentation (Gorriz et al., 2017). One notable contribution addressing data selection for segmentation is KnowWhat-ToLabel (Dawoud et al., 2023), which introduces a scoring function to construct a set of support samples for few-shot microscopy image cells segmentation. Although this method outperforms traditional scoring functions like Shannon's entropy and MC-dropout, it relies on model training to compute scores, which leads to high computational costs. Additionally, data selection remains underexplored in medical image segmentation tasks like polyp segmentation. As polyp segmentation continues to grow in importance for clinical practice, efficient strategies for curating high-quality datasets are increasingly necessary.

To address these limitations, to the best of our knowledge, we are the first to propose a *training-free dataset pruning* method, namely Pruning by Representation, Image-based Modeling, and Evaluation (*PRIME*) for polyp segmentation. Rather than relying on model-based metrics, *PRIME* constructs a similarity network to quantify the similarity among images in a target dataset. *PRIME* then exploits community detection (Blondel et al., 2008) (see Fig. 1) to retain a diverse and representative subset of the original dataset, thus effectively removing redundant data. Critically, our *PRIME* does not require model training, thus reducing the computational costs associated with data reduction. PRIME ultimately reduces annotation effort, enables faster segmentation model training, and reduces domain adaptation costs across medical centers. Our main contributions are as follows:

- **Training-Free Dataset Pruning:** We introduce a new, training-free dataset pruning method, *PRIME*, that first constructs a similarity network among images in the target dataset; then detects communities to select a much smaller, yet diverse and representative subset of the original data, thus eliminating redundant images.

- **Robust Generalizability Across Datasets:** By retaining a diverse and representative subset while pruning redundant images, our *PRIME* consistently achieves high DICE scores across multiple datasets and similarity metrics. This robustness underscores the broad applicability of *PRIME* to polyp segmentation.
- **Reduction in Annotation and Computational Costs:** Our *PRIME* achieves up to 56.2% data reduction, thus lowering the annotation effort while maintaining high segmentation performance with only a 0.5% drop in the DICE score. The training-free design of our pruning algorithm reduces overall computational costs and enables $2.3\times$ faster training of segmentation models compared to training in the entire dataset.

The remaining of the paper is organized as follows: Section 2 discusses related work on polyp segmentation and dataset pruning methods. Section 3 describes our method. Section 4 presents the experimental evaluations. Section 5 summarizes our main contributions.

## 2. Related Work

### 2.1. Polyp Segmentation

Polyp segmentation has been extensively studied in medical imaging due to its paramount importance in the diagnosis and prevention of colorectal cancer. Deep learning, particularly U-shaped convolutional neural networks (CNN) and vision transformers such as U-Net (Ronneberger et al., 2015), Attention UNet (Oktay et al., 2018), UNet++ (Zhou et al., 2018), UNet 3+ (Huang et al., 2020), DeepLabv3+ (Chen et al., 2018), PraNet (Fan et al., 2020), PolypPVT (Dong et al., 2021), CASCADE (Rahman and Marculescu, 2023a), SS-Former (Wang et al., 2022a), G-CASCADE (Rahman and Marculescu, 2024), and EMCAD (Rahman et al., 2024), have demonstrated remarkable performance for polyp segmentation. However, these models typically require large-scale pixel-level annotated datasets, which leads to substantial annotation costs. Furthermore, the variability in polyp size, shape, and appearance further exacerbates the need for diverse training data, which can be prohibitively expensive in terms of both labeling and computational resources.

### 2.2. Dataset Pruning and Data Selection

Dataset pruning and data selection aim to retain the most representative samples from a dataset while removing redundancies, an objective that is crucial for polyp segmentation due to the high cost of pixel-level annotations. Existing methods primarily target classification tasks, relying on training-based metrics or scalar scores like compactness (Castro et al., 2018; Yang et al., 2022), diversity (Aljundi et al., 2019), or forgetfulness (Toneva et al., 2018) to identify the most informative samples. Another line of work focuses on synthesizing smaller yet informative datasets via distillation (Such et al., 2020; Wang et al., 2018) or condensation (Zhao et al., 2021), however, they also involve complex training processes.

In the realm of semantic segmentation, especially medical image segmentation, dataset pruning remains comparatively underexplored. Most existing pruning or data selection methods have been developed and validated on classification benchmarks, with only a few adaptations to segmentation tasks. For instance, KnowWhatToLabel (Dawoud et al., 2023) introduces a consistency-based method to select training samples to reduce annotation costs in few-shot microscopy image cell segmentation. Nevertheless, this method still depends

on model training and has not been extended to polyp segmentation. The unique characteristics of polyp segmentation including the heterogeneous appearance of the polyp and the need for meticulous pixel-level labeling further emphasize the need for specialized and efficient data selection/pruning methods.

### 2.3. Existing Knowledge Gaps and the Need for a Training-Free Approach

Despite significant advances, a key knowledge gap persists: Existing solutions either require model training at some stage or focus on other domains without specifically addressing polyp segmentation. Given the high cost of annotations and logistical constraints in collecting and sharing medical data across institutions, computationally lightweight methods become crucial. Training-heavy pipelines exacerbate these challenges, particularly when dealing with large-scale, high-resolution datasets under strict privacy regulations.

We plan to address these limitations by introducing a *training-free dataset pruning* method designed explicitly for polyp segmentation. Rather than relying on gradient-based or generative scoring, we construct a similarity network among images and then apply community detection to isolate a diverse yet compact subset of the dataset. Our method not only substantially reduces the computational cost, but also alleviates the annotation burden, thus paving the way for faster and more cost-effective polyp segmentation workflows.

## 3. Method

In this section, we formally define the problem of dataset pruning and describe the two key components of our *PRIME*: similarity network construction and sample selection.

### 3.1. Dataset Pruning: Problem Definition

Let $\mathcal{D} = \{x_1, x_2, \ldots, x_n\}$ represent the original training dataset, where each $x_i$ is an image, and $n$ is the total number of images in the dataset. Our goal is to select a subset $\mathcal{D}' \subset \mathcal{D}$, such that $|\mathcal{D}'| = m$ and $m \ll n$, while preserving the generalizability of the model. Specifically, we want to minimize the performance difference between a model trained on the original dataset $\mathcal{D}$ and the pruned dataset $\mathcal{D}'$, i.e.:

$$\min_{\mathcal{D}'} \mathbb{E}_{x \in \mathcal{D}} \left[ \mathcal{L}(f_{\mathcal{D}}(x), f_{\mathcal{D}'}(x)) \right] \tag{1}$$

where $f_{\mathcal{D}}$ and $f_{\mathcal{D}'}$ represent models trained on the original dataset and the pruned dataset, respectively, and $\mathcal{L}$ denotes the loss function (e.g., cross-entropy or DICE). Our goal is to select $\mathcal{D}'$ such that it is significantly smaller than $\mathcal{D}$, but retains its diversity and representativeness, thus minimizing performance degradation compared to the dataset $\mathcal{D}$.

### 3.2. Similarity Network Construction

For efficient pruning, we first construct a similarity network $\mathcal{G} = (\mathcal{V}, \mathcal{E})$ from the training images, where each node $v_i \in \mathcal{V}$ represents an image $x_i \in \mathcal{D}$, and the edges $\mathcal{E}$ capture the similarity between image pairs. The similarity between any image pair $(x_i, x_j)$ is quantified using a metric such as structural similarity index (SSIM)[1] or Pearson's correlation coefficient (PCC). More precisely, the $SSIM \in [0, 1]$ of two images is calculated as in Eq. 2:

$$SSIM(x_i, x_j) = \frac{(2\mu_i \mu_j + C_1)(2\sigma_{ij} + C_2)}{(\mu_i^2 + \mu_j^2 + C_1)(\sigma_i^2 + \sigma_j^2 + C_2)} \tag{2}$$

---

1. SSIM similarity achieves slightly better pruning results than PCC as shown in Figs. 4 and 5.

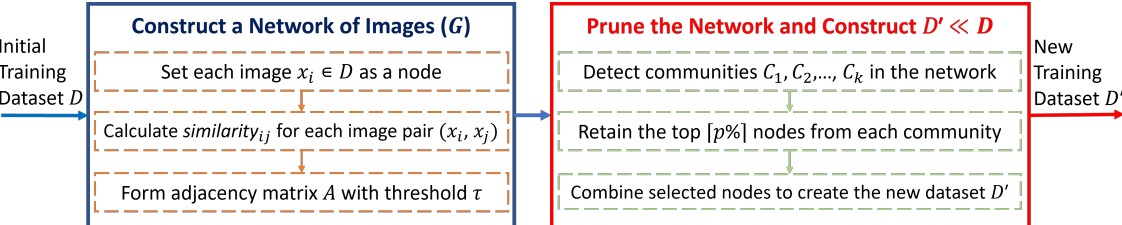

Figure 2: Proposed workflow diagram (left - network construction of images, right - pruning network to select a subset of images).

where $\mu_i$ and $\mu_j$ are mean intensities (average pixel values) of images $x_i$ and $x_j$, $\sigma_i^2$ and $\sigma_j^2$ are their variances, and $\sigma_{ij}$ is their covariance. Small positive constants $C_1$ and $C_2$ prevent instability in the division when variances or covariances are close to zero.

The $PCC \in [-1, 1]$ between a pair of images is computed as in Eq. 3:

$$PCC(x_i, x_j) = \frac{\sum (x_i - \bar{x}_i)(x_j - \bar{x}_j)}{\sqrt{\sum (x_i - \bar{x}_i)^2}\sqrt{\sum (x_j - \bar{x}_j)^2}} \tag{3}$$

where $\bar{x}_i$ and $\bar{x}_j$ are mean intensities (average pixel values) of images $x_i$ and $x_j$.

For each image pair $(x_i, x_j)$, an edge $e_{ij} \in \mathcal{E}$ is created between nodes $v_i$ and $v_j$ if their SSIM or PCC similarity exceeds a threshold $\tau$ (set based on the range of similarities in the target dataset). This results in an undirected graph $\mathcal{G}$, where pairs of images with sufficient SSIM or PCC similarities get connected. The adjacency matrix $\mathbf{A}$ of $\mathcal{G}$ is defined in Eq. 4:

$$\mathbf{A}_{ij} = \begin{cases} 1 & \text{if } similarity(x_i, x_j) \geq \tau, \\ 0 & \text{otherwise.} \end{cases} \tag{4}$$

Fig. 2 (left box) illustrates this process. For example, in Fig. 1, the similarity network of the CVC-ClinicDB dataset is shown for an SSIM threshold of $\tau = 0.92$; this network has 550 nodes and 1510 edges. More details on this network are provided in the Appendix B.

### 3.3. Sample Selection via Community Detection

Our pruning method exploits community detection (Blondel et al., 2008) within the similarity network $\mathcal{G}$. Communities in the network correspond to clusters of highly similar images (Fig. 1) which can help us to select a much smaller, yet representative subset from each community. The steps of our pruning method (Fig. 2, right box) are described next.

#### 3.3.1. COMMUNITY DETECTION

We use a community detection algorithm, namely the Louvain method (Blondel et al., 2008), to identify communities in the similarity network. Our goal is to maximize the modularity $Q$ of the network, defined as in Eq. 5:

$$Q = \frac{1}{2|\mathcal{E}|} \sum_{i,j} \left[ A_{ij} - \frac{k_i k_j}{2|\mathcal{E}|} \right] \delta(c_i, c_j) \tag{5}$$

where $|\mathcal{E}|$ is the cardinality of edges $\mathcal{E}$, $A_{ij}$ denotes adjacency as in Eq. 4 (1 if nodes $i$ and $j$ are connected, 0 otherwise), $k_i$ and $k_j$ are the node degrees of nodes $i$ and $j$, and $c_i$ and $c_j$ indicate the communities of nodes $i$ and $j$. The indicator $\delta(c_i, c_j) = 1$ if $c_i = c_j$ (same community), and 0 otherwise. This process identifies the network communities[2] $\mathcal{C}_1, \mathcal{C}_2, \ldots, \mathcal{C}_k$, where each community $\mathcal{C}_k$ consists of a set of highly similar images.

2. We note that, in networks with a weaker structure, these groupings may be less distinct; this issue can be mitigated by using a smaller similarity threshold $\tau$.

### 3.3.2. Sample selection

For each detected community $\mathcal{C}_k$, we retain only a representative subset of nodes (i.e., images) based on the size of the community: *i)* If $|\mathcal{C}_k| = 1$, we retain the single node, as it represents a unique image in the dataset; *ii)* If $|\mathcal{C}_k| > 1$, we retain the top $\lceil p\% \rceil$ of nodes based on their (higher) node degree within the community; the remaining nodes in the community are pruned. Finally, the retained subset $\mathcal{D}'$ can be represented as in Eq. 6:

$$\mathcal{D}' = \bigcup_{k=1}^{K} \mathcal{C}_k^{\lceil p\% \rceil} \tag{6}$$

where $\mathcal{C}_k^{\lceil p\% \rceil}$ represents the top $\lceil p\% \rceil$ nodes selected from community $\mathcal{C}_k$, and $K$ is the total number of detected communities.

Intuitively, the similarity threshold $\tau$ controls the network density, with lower values forming fewer, but denser communities and thus enabling higher pruning. A smaller $\lceil p\% \rceil$ retains fewer nodes per community, which ensures that each community is represented by its most representative images, thus reducing redundancy while preserving diversity.

## 4. Experiments

### 4.1. Segmentation Network Architectures

We use the PVT-v2-b2 (PVT) (Wang et al., 2022b) and ResNet50 (R50) (He et al., 2016) encoders (which are hierarchical backbones) and extract features from four stages. Then, we use the CASCADE decoder[3] (Rahman and Marculescu, 2023b) (which is a local attention-based cascaded decoder) to decode and obtain the segmentation outputs of different stages. Finally, we use the output from the last stage to obtain the final segmentation map. We adopt the multi-stage loss aggregation for training as in (Rahman and Marculescu, 2023b).

### 4.2. Implementation Details

We implement all our experiments in Pytorch 1.11.0 and train all models on a single NVIDIA RTX A6000 GPU with 48GB of memory. We use different similarity thresholds ($\tau$) to construct the similarity network. We do not use any data augmentations in our experiments.

We use the AdamW optimizer (Loshchilov and Hutter, 2017) with a learning rate and weight decay of 1e-4 in our experiments. We use the combined weighted IoU and weighted Binary Cross Entropy (BCE) loss function for the polyp segmentation on CVC-ClincDB and Kvasir datasets. We train the model for 200 epochs with a batch size of 16. We also utilize the pre-trained weights on ImageNet for backbone networks. Finally, we report the average DICE score (%) over five runs for evaluation.

### 4.3. Results

**Impact of $\lceil p\% \rceil$ community representative:** We conducted an ablation study on CVC-ClinicDB dataset with the PVT-CASCADE model to see the impact of selecting different percentages of representatives from each community (Fig. 3). We can conclude that the DICE score increases only marginally when we select more than $\lceil 10\% \rceil$ from each community. Hence, we select the top $\lceil 10\% \rceil$ samples from each community for model training purposes and prune/remove the rest of the community nodes in all of our experiments.

---

3. Other decoders such as EMCAD (Rahman et al., 2024) can be also used as shown in Appendix C.3.

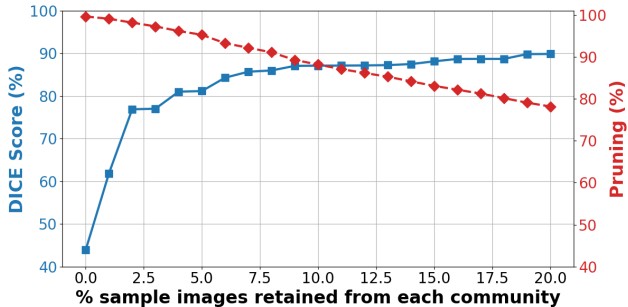

Figure 3: Pruning (%) and DICE scores (%) vs. images (%) selected from each community in CVC-ClinicDB dataset (total of 550 images) with PVT-CASCADE model. We use SSIM with a median threshold $\tau$=0.5 for a denser network construction with fewer communities.

Table 1: Experimental results (DICE %) on CVC-ClinicDB dataset using PVT-CASCADE and ResNet50 (R50)-CASCADE. $\tau$ is a similarity threshold that controls the density (i.e., pruning rate) of the similarity network. We select top $\lceil 10\% \rceil$ images based on the degree of each community in SSIM-based similarity networks.

| Architectures | Methods/($\tau$, pruning %) | ($\tau$=0.92, 55.5%) | ($\tau$=0.95, 31.3%) | ($\tau$=0.97, 15.1%) | ($\tau$=1, 0% pruning) |
|---|---|---|---|---|---|
| PVT-CASCADE | Entropy (Shannon, 1948) | 89.82±1.4 | 91.77±1.0 | 92.63±0.8 | |
| PVT-CASCADE | MC-dropout (Gal and Ghahramani, 2016) | 90.18±1.3 | 92.25±0.9 | 92.81±0.7 | |
| PVT-CASCADE | KnowWhatToLabel (Dawoud et al., 2023) | 90.34±1.4 | 92.62±0.9 | 92.87±0.8 | 94.29±0.5 |
| PVT-CASCADE | Random | 89.94±3.3 | 91.99±2.1 | 92.24±1.3 | |
| PVT-CASCADE | **PRIME (Ours)** | **92.85±1.3** | **94.18±0.7** | **94.48±0.5** | |
| R50-CASCADE | Entropy (Shannon, 1948) | 89.37±1.5 | 91.43±0.9 | 92.21±0.7 | |
| R50-CASCADE | MC-dropout (Gal and Ghahramani, 2016) | 89.96±1.3 | 91.94±0.8 | 92.42±0.7 | |
| R50-CASCADE | KnowWhatToLabel (Dawoud et al., 2023) | 90.10±1.3 | 92.29±0.9 | 92.51±0.6 | 93.97±0.4 |
| R50-CASCADE | Random | 89.53±3.1 | 91.61±1.9 | 91.86±1.4 | |
| R50-CASCADE | **PRIME (Ours)** | **92.45±1.1** | **93.82±0.8** | **94.08±0.4** | |

**Generalizability in multiple models:** Table 1 shows the effectiveness of our *PRIME* in improving generalizability across the PVT-CASCADE and R50-CASCADE models. At a similarity threshold of $\tau$=0.92 (i.e., 55.5% pruning), PVT-CASCADE achieves a DICE score of 92.85% vs. 89.94% with random selection, and R50-CASCADE scores 92.45% compared to 89.53%. As pruning decreases, our *PRIME* consistently outperforms the random selection. *PRIME* also outperforms KnowWhatToLabel, Entropy, and MC-dropout pruning methods. Even at $\tau$=0.97 (15.1% pruning), our *PRIME* maintains high DICE scores in both models, closely matching the results obtained with the full dataset. This demonstrates our *PRIME*'s ability to preserve data diversity and ensure robust segmentation performance with significantly reduced data. More results on Kvasir dataset are shown in Appendix C.2.

**Generalizability on multiple datasets (centers) and similarity metrics:** Figs. 4 and 5 show the efficacy of our *PRIME* on the CVC-ClinicDB and Kvasir datasets, using both PCC and SSIM similarity metrics. Our *PRIME* consistently yields higher DICE scores than random pruning, even at higher pruning rates. For instance, at 31.3% pruning on the CVC-ClinicDB (Fig. 4), our PCC-based pruning achieves 94.02% DICE score, and SSIM-based pruning reaches 94.2%, compared to only 92.0% with random pruning. At 71.9% pruning on the Kvasir dataset (Fig. 5), our *PRIME* maintains DICE scores of 91.1% (PCC) and 91.6% (SSIM), while random pruning drops to 89.6%. Our *PRIME* also consistently outperforms KnowWhatToLabel, Entropy, and MC-dropout pruning methods. Finally, the training time per epoch decreases from about 100 secs at 0% pruning to 28 secs at 71.9% pruning, thus validating the efficiency and robustness of our *PRIME* across datasets or imaging centers.

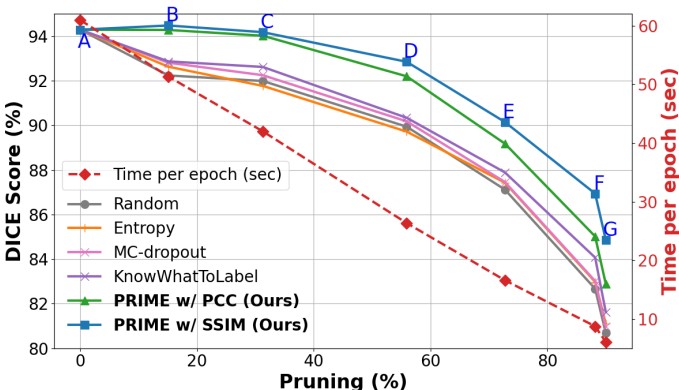

Figure 4: Pruning (%) vs. DICE (left axis) and Time (right axis) on the CVC-ClinicDB dataset (Bernal et al., 2015) (550 images) with the PVT-CASCADE model. SSIM thresholds $\tau=$[1, 0.97, 0.95, 0.92, 0.9, 0.88, 0.85] are used to construct similarity networks, while PCC thresholds are adjusted to achieve similar pruning rates. Training time per epoch is reported averaging over 200 epochs. Our *PRIME* prunes 55.5%[(D)] of data with only a 1.4%[(A-D)] drop in DICE and reduces 2.3× training time compared to training on the entire dataset. Notably, we can prune 15%[(B)] of images with a 0.2%[(B-A)] increase in DICE.

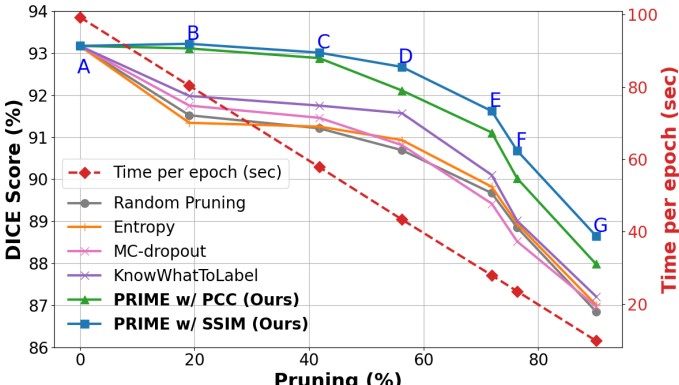

Figure 5: Pruning (%) vs. DICE score (left axis) and Time (right axis) on the Kvasir dataset (Jha et al., 2020) (900 images) with the PVT-CASCADE model. Training time per epoch is reported averaging over 200 epochs. SSIM thresholds $\tau=$[1, 0.97, 0.90, 0.85, 0.8, 0.75, 0.7] are used to construct similarity networks, while PCC thresholds are adjusted to achieve similar pruning rates. Our *PRIME* prunes 56.2%[(D)] of data with only a 0.5%[(A-D)] drop in DICE and achieves 2.3× faster training compared to training on the entire dataset.

**Scalability on large video polyp segmentation dataset:** To demonstrate the scalability of our PRIME method, we evaluate it on SUN-SEG (Table 2), a large-scale video polyp segmentation dataset with 19,544 training images, significantly larger than standard polyp datasets. Handling such large datasets is computationally expensive and increases annotation costs, making dataset pruning crucial for practical deployment. Our method effectively reduces dataset size by 69.7% while maintaining segmentation performance comparable to training on the full dataset. PRIME (SSIM) achieves less than a 1% drop in DICE scores on seen test sets, whereas random pruning results in up to a 3.13% performance drop, demonstrating that our structured community-based selection strategy retains essential information while significantly reducing data redundancy. In addition, PRIME reduces storage and computational costs, making large-scale dataset management more ef-

Table 2: Results on a large video polyp dataset (SUN-SEG) (Ji et al., 2022). We use the PVT-CASCADE network and run each model for 30 epochs. Our PRIME prunes 69.7% of the data with an SSIM similarity threshold of 0.7, while PCC thresholds are adjusted to achieve similar pruning rates. We report the average DICE score (%) over three runs.

| Pruning Method | Easy Seen (%) | Easy Unseen (%) | Hard Seen (%) | Hard Unseen (%) |
|---|---|---|---|---|
| Full Dataset | 92.47 | 80.65 | 87.77 | 80.80 |
| PRIME w/ SSIM (**Ours**) | 91.55 | 80.06 | 86.95 | 80.51 |
| PRIME w/ PCC (**Ours**) | 91.12 | 79.43 | 86.64 | 79.86 |
| Random Pruning | 90.23 | 77.86 | 85.34 | 77.67 |

Table 3: Effects of augmentation on training PVT-CASCADE model using the full dataset, our PRIME, and random pruning. We apply random rotation and flips as augmentations. We report the average DICE score (%) over three runs.

| Dataset | Training Data | No Augmentation | With Augmentation | Improvement |
|---|---|---|---|---|
| CVC-ClinicDB | Full Dataset | 94.29% | 94.63% | +0.34% |
| | PRIME w/ SSIM (Ours) | 92.85% | 93.72% | +0.87% |
| | Random Pruned | 89.94% | 90.76% | +0.82% |
| SUN-SEG | Full Dataset | 92.47% | 92.91% | +0.44% |
| | PRIME w/ SSIM (Ours) | 91.55% | 92.57% | +1.02% |
| | Random Pruned | 90.23% | 90.98% | +0.75% |

ficient without sacrificing segmentation quality. These results confirm that PRIME is a scalable, cost-effective solution for large medical imaging datasets.

**Generalizability on unseen testset:** To assess the generalizability of our PRIME method, we evaluate segmentation performance on the unseen testsets from SUN-SEG (Table 2). Despite pruning 69.7% of the dataset, PRIME (SSIM) maintains segmentation DICE scores within 0.6% of the full dataset, demonstrating that our method preserves diverse and informative samples essential for robust model learning. In contrast, random pruning leads to a 3.13% performance drop, indicating that naive selection disrupts essential feature distribution, negatively affecting generalization. These results confirm that PRIME effectively balances dataset reduction with performance retention, thus ensuring strong generalization to unseen cases while significantly reducing annotation cost.

**Effect of augmentation during training:** As shown in Table 3, augmentation has minimal impact ($<0.5\%$) when using the full dataset. However, for PRIME pruned datasets, augmentation improves the DICE scores by 0.87–1.02%, suggesting that augmentations compensate for reduced dataset size.

## 5. Conclusion and Future Work

In this paper, we have introduced *PRIME*, a training-free dataset pruning method to minimize image annotation (labeling) efforts and enable efficient training of segmentation models. Experiments on multiple medical image segmentation datasets show its potential to maintain high DICE scores while reducing computational and data annotation costs.

Future work will focus on developing more advanced metrics with critical shape information, a deeper analysis of similarity networks, and expanding experiments to diverse datasets (including 3D segmentation). Our *PRIME* holds promise in accelerating research in continual learning, active learning, contrastive learning, and few-shot learning by enhancing data efficiency in resource-intensive applications.

## Acknowledgments

This work is supported in part by the NSF grant CNS 2007284, in part by the iMAGiNE Consortium (https://imagine.utexas.edu/), and in part by the Texas Health Catalyst award.

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

## Appendix A. Datasets Details

CVC-ClinicDB (Bernal et al., 2015) contains 612 images, which are extracted from 31 colonoscopy videos. Kvasir includes 1,000 polyp images, which are collected from the polyp class in the Kvasir-SEG dataset (Jha et al., 2020). Following the settings in CASCADE (Rahman and Marculescu, 2023b), we adopt the same 550 and 900 images from CVC-ClinicDB and Kvasir datasets as the training set, and the remaining 62 and 100 images, respectively, are used as testsets.

The SUN-SEG dataset (Ji et al., 2022; Misawa et al., 2021) is a large-scale video polyp segmentation benchmark consisting of 49,136 polyp frames and 109,554 non-polyp frames, with 19,544 training images and 29,592 test images. It provides diverse annotations, including pixel-wise masks, boundaries, scribbles, and polygons, supporting both fully and weakly supervised learning. The test set is further categorized into easy and hard cases, with 4,719 easy seen, 12,351 easy unseen, 3,882 hard seen, and 8,640 hard unseen images, ensuring a rigorous evaluation of model generalization. As one of the most well-annotated and high-quality polyp segmentation datasets, SUN-SEG serves as a critical benchmark for real-world endoscopic applications.

## Appendix B. Detailed Analysis of Similarity Network Properties and Community Detection

The structural properties of the constructed similarity network (900 nodes, 8922 edges) in Fig. 6 provide compelling evidence for the utility of community detection in training-free dataset pruning. Below, we analyze key metrics and their implications:

### B.1. Network Connectivity and Cohesion

- **Average Degree (19.827):** Each node is connected to 20 others, on average, indicating robust pairwise similarity relationships. This density ensures that communities are well-anchored by hubs (high-degree nodes) while retaining fine-grained connections between niche samples.

- **Graph Density (0.022):** The sparsity of the network (only 2.2% of possible edges exist) reflects a carefully calibrated similarity threshold, filtering out weak or noisy relationships. This sparsity enhances the discriminative power of detected communities, as retained edges likely correspond to semantically meaningful similarities.

### B.2. Small-World Characteristics

- **Average Path Length (3.348):** The short average distance between nodes (3.348 hops) suggests a small-world topology, where tightly knit communities are interconnected by a few bridging nodes (Watts and Strogatz, 1998). This property enables efficient traversal of the network during pruning, thus ensuring that representative samples can be selected without exhaustive search.

- **Network Diameter (10):** The longest shortest path spans 10 edges, indicating that even the most dissimilar images are relatively proximate in the feature space. This

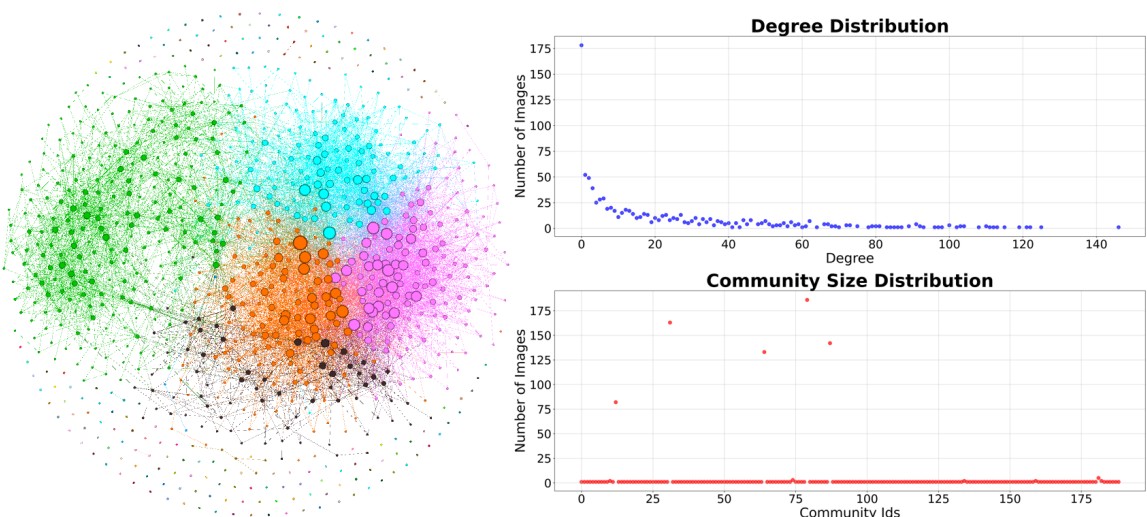

Figure 6: Communities identified with Structural Similarity Index (SSIM) threshold of 0.80 in the Kvasir dataset (Jha et al., 2020). Gephi's community detection (Blondel et al., 2008) identifies 189 communities with a modularity of 0.512. The degree distribution shows the number of nodes with varying connectivity (0 to >175, left-skewed), while the community distribution highlights the number of nodes per community (0 to >175). By considering top ⌈10%⌉ nodes from each community, we select only 253 (28.1%) representative nodes and prune the rest of the nodes (71.9%).

compactness supports the hypothesis that the dataset contains latent hierarchical structures resolvable via community detection.

### B.3. Community Detection Efficacy

- **Modularity (0.512):** A modularity score >0.3 confirms the statistically significant community structure. The modularity value of 0.512 indicates strong separation between groups, where intra-community edges significantly outnumber inter-community edges. This ensures that the detected clusters are cohesive and distinct, aligning with visually or pathologically meaningful subgroups.

- **Connected Components (184 → 189 Communities):** The network initially contains 184 isolated components, but community detection resolves 189 clusters, demonstrating that the Gephi's algorithm (Blondel et al., 2008) successfully identifies substructures within connected components. This granularity is critical for capturing fine-grained variations (e.g., polyp subtypes or imaging artifacts).

### B.4. Local Clustering and Redundancy Reduction

- **Average Clustering Coefficient (0.597):** The high clustering coefficient indicates that the nodes tend to form tightly connected triads, a hallmark of homophilic networks where similar nodes cluster together (Watts and Strogatz, 1998). This property ensures communities are internally homogeneous, reducing redundancy and enabling the selection of representative samples without oversampling.

- **Hub-Driven Cohesion:** Hubs (high-degree nodes) act as central coordinators, linking disparate regions of the network. By prioritizing hubs during pruning, our method retains images that anchor multiple communities, thus preserving the global dataset structure while minimizing information loss.

### B.5. Implications for Training-Free Dataset Pruning

The network's small-world compactness (short average path of 3.348), scale-free topology (evidenced by hubs with degree >100 and average degree 19.827), and strong modularity (0.512) collectively validate community detection as a principled framework for dataset pruning. By leveraging these properties, our *PRIME* achieves:

- *Efficiency:* Short paths and hierarchical communities reduce computational overhead.

- *Representativeness:* Cohesive clusters preserve clinical diversity.

- *Interpretability:* Communities align with domain-specific patterns (e.g., pathology, imaging protocols).

Our method is particularly advantageous for polyp segmentation, where resource limitations (e.g., expert humans available to do annotation or computing resources to run intense computations) demand strategies that balance performance, efficiency, and clinical relevance.

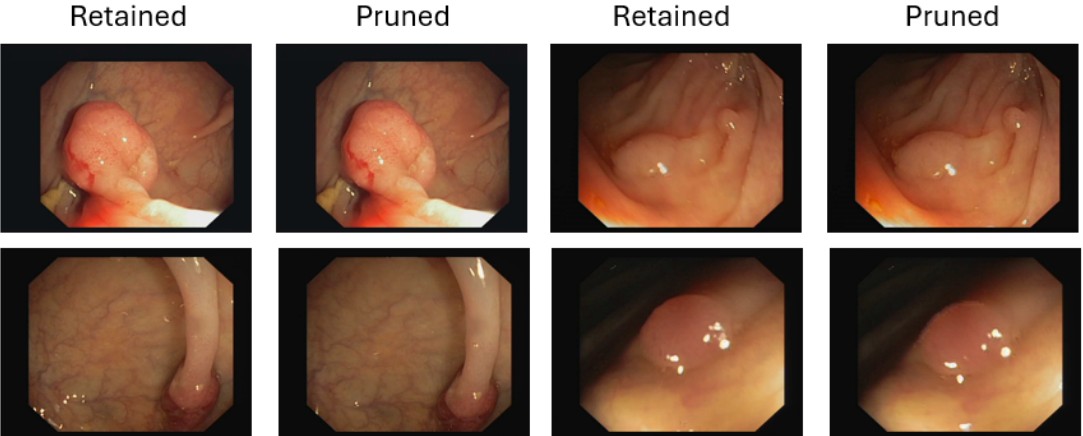

Figure 7: Our PRIME retained and pruned samples from different communities in CVC-ClinicDB dataset (pruning around 15% highly similar images). Although more than one image can be retained or pruned depending on the community size, here we report pairs of retained and pruned samples from the same community.

## Appendix C.  Additional Experiments

### C.1. Visualization of retained and pruned samples from different communities

To further illustrate how our PRIME pruning method retains diverse and representative samples while removing redundant ones, we present a qualitative analysis in Figure 7. This

Table 4: Experimental results (DICE %) on Kvasir dataset (Jha et al., 2020) using PVT-CASCADE and ResNet50 (R50)-CASCADE. $\tau$ is a similarity threshold that controls the density (i.e., pruning rate) of the similarity network. We select top $\lceil 10\% \rceil$ images based on the degree of each community in SSIM-based similarity networks.

| Architectures | Methods/($\tau$, pruning %) | ($\tau$=0.83, 56.2%) | ($\tau$=0.85, 41.8%) | ($\tau$=0.88, 19.1%) | ($\tau$=1, 0% pruning) |
|---|---|---|---|---|---|
| PVT-CASCADE | Entropy (Shannon, 1948) | 90.93±1.2 | 91.25±0.9 | 91.34±0.6 | |
| PVT-CASCADE | MC-dropout (Gal and Ghahramani, 2016) | 90.81±1.3 | 91.46±0.9 | 91.75±0.7 | 93.17±0.4 |
| PVT-CASCADE | KnowWhatToLabel (Dawoud et al., 2023) | 91.57±1.3 | 91.75±0.7 | 91.98±0.7 | |
| PVT-CASCADE | Random | 90.69±2.8 | 91.21±1.9 | 91.52±1.2 | |
| PVT-CASCADE | **PRIME (Ours)** | **92.67±1.1** | **93.01±0.6** | **93.22±0.5** | |
| R50-CASCADE | Entropy (Shannon, 1948) | 90.58±1.4 | 91.06±1.0 | 91.22±0.7 | |
| R50-CASCADE | MC-dropout (Gal and Ghahramani, 2016) | 90.63±1.2 | 91.41±0.9 | 91.62±0.6 | 93.02±0.5 |
| R50-CASCADE | KnowWhatToLabel (Dawoud et al., 2023) | 91.41±1.4 | 91.62±0.8 | 91.84±0.7 | |
| R50-CASCADE | Random | 90.53±2.9 | 91.12±1.7 | 91.45±1.5 | |
| R50-CASCADE | **PRIME (Ours)** | **92.01±1.3** | **92.73±0.9** | **92.96±0.6** | |

figure presents examples of retained and pruned images from different communities in the CVC-ClinicDB dataset, where approximately 15% of highly similar images were removed.

Our pruning strategy ensures that visually similar, yet redundant, images are removed while preserving the most informative and diverse samples within each community. As seen in Figure 7, the retained images effectively represent polyp characteristics, mucosal textures, and variations in lighting, while the pruned images contain nearly identical structural features with minimal additional information. By eliminating these redundant samples, our method reduces annotation costs without compromising dataset diversity, leading to improved segmentation performance.

Although multiple images can be retained or pruned depending on the community size, for clarity, we present pairs of retained and pruned samples from the same community. This visualization reinforces that PRIME does not randomly discard images, but rather strategically selects a subset that maximally preserves diversity while minimizing redundancy.

## C.2. Results on the Kvasir dataset

Table 4 shows the efficacy of *PRIME* in balancing dataset pruning and segmentation performance. While existing methods (e.g., entropy-based selection, MC-dropout) exhibit performance degradation under aggressive pruning (e.g., 56.2% reduction at $\tau = 0.83$), *PRIME* consistently outperforms baselines, achieving higher DICE scores with narrower standard deviations (e.g., **92.11% vs. 90.93%** for PVT-CASCADE at $\tau = 0.83$).

Notably, at $\tau = 0.88$ (19.1% pruning), *PRIME* nearly matches the full-dataset baseline (93.15% vs. 93.17%), highlighting its ability to retain critical samples through community detection in similarity networks. This contrasts with random pruning, which suffers from high variance (e.g., ±2.8 at $\tau = 0.83$), emphasizing the non-triviality of sample selection.

Our PRIME's robustness across architectures (PVT vs. ResNet50) further validates its generalizability, though PVT's superior performance suggests architectural advantages in capturing polyp features. These findings position *PRIME* as a computationally efficient alternative for resource-constrained medical imaging tasks.

## C.3. Results of ClinicDB dataset with the EMCAD decoder:

Table 5 shows that *PRIME* effectively optimizes data efficiency for polyp segmentation using recent EMCAD decoder (Rahman et al., 2024) as well, particularly in data-rich regimes. Under aggressive pruning ($\tau = 0.92$, 55.5%), *PRIME* achieves **92.87% DICE** for PVT-EMCAD, outperforming entropy-based selection by **+3.14%** and random pruning by **+3.02%**, despite higher variability ($\pm 1.5$ vs. $\pm 3.6$ for random). This underscores its ability to retain diagnostically critical samples even with significant dataset reductions. As pruning relaxes ($\tau = 0.97$, 15.1%), *PRIME*'s performance nears the full-dataset baseline (94.61% vs. 94.65% for PVT-EMCAD), suggesting diminishing returns for retaining additional data. Notably, PVT-EMCAD consistently surpasses ResNet50-EMCAD (e.g., **94.61% vs. 94.15%** at $\tau = 0.97$), likely due to its hierarchical attention mechanisms better capturing polyp boundaries. These insights position *PRIME* as a architecture-agnostic dataset pruning method.

Table 5: Experimental results (DICE %) on CVC-ClinicDB dataset using PVT-EMCAD and ResNet50 (R50)-EMCAD. $\tau$ controls the similarity threshold and pruning rate. SSIM similarity achieves better DICE score than PCC in similar pruning rate.

| Architectures | Methods/($\tau$, pruning %) | ($\tau$=0.92, 55.5%) | ($\tau$=0.95, 31.3%) | ($\tau$=0.97, 15.1%) | ($\tau$=1, 0% pruning) |
|---|---|---|---|---|---|
| PVT-EMCAD | Entropy (Shannon, 1948) | 89.73±1.7 | 91.83±1.4 | 92.85±1.2 | |
| PVT-EMCAD | MC-dropout (Gal and Ghahramani, 2016) | 90.15±1.6 | 92.31±1.3 | 93.05±1.1 | 94.65±0.6 |
| PVT-EMCAD | KnowWhatToLabel (Dawoud et al., 2023) | 90.36±1.7 | 92.81±1.1 | 93.14±1.0 | |
| PVT-EMCAD | Random | 89.85±3.6 | 91.87±2.4 | 92.43±1.7 | |
| PVT-EMCAD | **PRIME w/ PCC (Ours)** | **92.58±1.4** | **94.23±0.9** | **94.34±0.7** | |
| PVT-EMCAD | **PRIME w/ SSIM (Ours)** | **92.87±1.5** | **94.47±1.0** | **94.61±0.8** | |
| R50-EMCAD | Entropy (Shannon, 1948) | 89.31±1.9 | 91.51±1.5 | 92.25±1.3 | |
| R50-EMCAD | MC-dropout (Gal and Ghahramani, 2016) | 89.90±1.7 | 92.01±1.2 | 92.49±1.1 | 94.26±0.5 |
| R50-EMCAD | KnowWhatToLabel (Dawoud et al., 2023) | 90.14±1.7 | 92.39±1.1 | 92.62±1.0 | |
| R50-EMCAD | Random | 89.46±3.5 | 91.72±2.3 | 91.97±1.8 | |
| R50-EMCAD | **PRIME w/ PCC (Ours)** | **92.25±1.5** | **93.72±0.9** | **94.02±0.6** | |
| R50-EMCAD | **PRIME w/ SSIM (Ours)** | **92.41±1.4** | **93.91±1.0** | **94.15±0.7** | |

# Appendix D. Computational Complexity of PRIME

## D.1. Theoretical Computational Complexity Analysis

The computational complexity of our PRIME is primarily driven by similarity matrix construction, which involves computing pairwise similarities for a dataset of $N$ images. This step requires $N(N-1)/2$ comparisons, leading to an overall complexity of $O(N^2HW)$, where $H \times W$ represents image dimensions. Given the quadratic scaling, computing the similarity matrix for large datasets is computationally demanding, but highly parallelizable, thus allowing multi-GPU acceleration to make it feasible even at large scales.

The Louvain community detection algorithm, used to select representative samples from the similarity graph, is significantly more efficient than similarity computation, operating in $O(N \log N)$ for sparse graphs. Since our similarity networks are inherently sparse, the computational overhead of this step remains minimal even for large datasets. Additionally, as dataset pruning is a one-time preprocessing step, its cost is offset by the substantial

Table 6: Computational time for similarity matrix construction across datasets of increasing size. Here, we report the time (s) needed to preprocess (data loading, grayscale conversion, and resize to $352 \times 352$), and $N(N-1)/2$ pair-wise similarity metrics computation (N is the number of images in a dataset) in NVIDIA RTX A6000 (Ada) GPUs with 48GB memory.

| Dataset | SSIM (1 GPU) (s) | SSIM (8 GPUs) (s) | PCC (1 GPU) (s) |
|---|---|---|---|
| CVC-ClinicDB (550) | 73.17 | 37 | 8.01 |
| Kvasir (900) | 190.55 | 72 | 15.80 |
| SUN-SEG (19,544) | 89,852.13 ( 25 hrs) | 10,920 ( 3 hrs) | 563.85 |

Table 7: Comparison of PRIME with fixed-iteration random sampling.

| Performance Metric | PRIME (Pruned Training) | Fixed-Iteration Random Sampling |
|---|---|---|
| **Storage Cost** | Reduced (Only pruned dataset stored) | Full dataset stored |
| **Annotation Cost** | Reduced (Only pruned subset labeled) | Full dataset annotated |
| **Training Speed** | Comparable (Reduced dataset) | Comparable (but full dataset overhead) |
| **DICE Score** | Minimal drop ($<0.5\%$) | Minimal to no drop |

savings in storage and annotation efforts, making PRIME a scalable and efficient solution for large-scale medical image curation.

### D.2. Empirical Computational Efficiency

To validate the scalability of PRIME, we measure the similarity matrix computation time on datasets of increasing sizes. Table 6 summarizes our findings, comparing the computational costs of SSIM on single and multi-GPU setups and PCC on a single GPU.

Table 6 demonstrates that multi-GPU acceleration significantly reduces computational overhead, enabling SSIM-based similarity computation for SUN-SEG (19,544 images) in 3 hours on 8 GPUs compared to 25 hours on a single GPU. In addition, the PCC similarity matrix can be computed within only 563.85 sec (562 sec preprocessing and 1.85 sec similarity matrix computation) on a single GPU. This ensures that dataset pruning remains computationally feasible even at large scales, making PRIME a practical solution to reduce annotation costs and storage burdens in medical imaging.

While similarity computation follows quadratic scaling, our results confirm that multi-GPU parallelization enables PRIME to efficiently process large datasets. Future optimizations, such as batch comparisons, could further extend the scalability to datasets containing millions of images, strengthening PRIME's role as an effective cost-saving and storage-efficient solution for large-scale medical image curation.

## Appendix E. PRIME vs. Fixed-Iteration Random Sampling

The fixed iterations per epoch can achieve comparable training speed to our PRIME-pruned dataset. However, our primary goal is not just faster training, but also reducing annotation and storage costs, with training efficiency being a byproduct of dataset pruning.

Unlike random sampling, which selects a subset of images per iteration, but still requires storing and accessing the entire dataset, PRIME systematically prunes redundant images while preserving dataset diversity, thus reducing annotation effort without compromising

segmentation performance. This distinction is crucial in medical imaging, where expert annotation is costly. Even with fixed-iteration random sampling, the need to maintain and retrieve the full dataset creates unnecessary storage and computational overhead, whereas PRIME significantly reduces both. Table 7 provides a structured comparison between PRIME and fixed-iteration random sampling.

## Appendix F. Current Limitations and Future Work

Our dataset pruning method balances data reduction with high segmentation accuracy across models, datasets, and similarity metrics. Additionally, as a training-free method, our pruning method minimizes the annotation costs and effort by effectively identifying representative samples, which is particularly advantageous in medical imaging where data annotation (labeling) is time-intensive and costly.

However, the DICE scores decline at extreme pruning levels, indicating a potential loss of critical data diversity. Furthermore, the method requires similarity threshold tuning, which varies by dataset. Future work will focus on exploring more advanced metrics (such as shape-aware and boundary-sensitive similarity measures), extending our method to other medical imaging tasks, and incorporating adaptive similarity thresholding to enhance robustness and generalizability across diverse medical imaging scenarios.

