# OpenReview forum: "Training-Free Dataset Pruning for Polyp Segmentation via Community Detection in Similarity Networks"
_MIDL.io/2025/Conference — MIDL 2025 Oral_

### Official Review · Reviewer_tCsW · 2025-02-09

**Confidence:** 4
**Preliminary Rating:** 4
**Recommendation:** Poster
**Final Rating:** 4

**Summary:**

This paper proposes a novel training-free dataset pruning method. The main contribution lies in utilizing community algorithms to identify and remove redundant images from the dataset. Experiments demonstrate that the proposed PRIME algorithm achieves good results on two polyp segmentation datasets.

**Strengths:**

1. The task of pruning medical image segmentation datasets is rarely discussed, and this paper helps inspire future research on the topic.
2. The proposed training-free pruning method is interesting.
3. The proposed method achieves better performance than existing dataset pruning methods.

**Weaknesses:**

1. The efficiency of the proposed method needs to be reported. How much time is required to complete the dataset pruning process?
2. It is necessary to supplement the analysis with specific polyp images to discuss why the deleted images are "less important" compared to the retained ones. This is particularly relevant to Figure 4, where the proposed method achieves even higher Dice scores after deleting around 20% of the samples.
3. On page 12, the test set size of CVC-ClinicDB should be 62 instead of 50.

**Detailed Comments:**

In the rebuttal phase, it is suggested to add additional analyses on algorithm efficiency and sample visualisation.

**Justification Of The Final Rating:**

After reading the comments of the other reviewers and the author's response, I think the problems with this paper have basically been properly addressed. Therefore, I will keep my score as ‘weak accept’.

**Justification Of The Preliminary Rating:**

Overall, the advantages of this paper outweigh the weaknesses, and the proposed method is helpful for building data-efficient deep learning models. Therefore, my current recommendation is weak accept.

**Questions To Address In The Rebuttal:**

See weakness and detailed comments.

---

> ### Author Response · Authors · 2025-03-08
>
> We sincerely thank the reviewer for the constructive feedback, and highlighting both the strengths and areas for improvement in our work. Below, we address each concern raised by the reviewer and outline how we incorporate these suggestions in our revised manuscript.
>
> ### **Q1. The efficiency of the proposed method needs to be reported. How much time is required to complete the dataset pruning process?**
>
> **Response:** We appreciate the reviewer’s request for a detailed efficiency analysis of our method. The dataset pruning process consists of two key computational steps:
>
> 1. **Similarity Matrix Construction** – This involves computing N(N-1)/2 pairwise similarities for all images, which scales as **O(N²)**. The time required depends on both the **hardware setup** and the chosen similarity metric.
>
> 2. **Community Detection (Node Selection)** – The **Louvain algorithm**, which operates in **O(N log N)** time, is used for selecting a diverse subset. This step remains **computationally efficient even for large datasets**.
>
> To provide a clearer understanding, we present the measured pruning times across different datasets and computational setups in Table R6 below:
>
> **Table R6:** Similarity matrix construction time in small to large-scale datasets. Here, we report the time (s) needed to preprocess (data loading, grayscale conversion, and resize to 352 × 352), and N(N-1)/2 (where N is the number of images in a dataset) pair-wise similarity metrics computation in NVIDIA RTX A6000 (Ada) GPUs with 48GB of memory.
>
> | **Dataset (#Images)**          | **SSIM (1 GPU) (s)**  | **SSIM (8 GPUs) (s)**  | **PCC (1 GPU) (s)** |
> |----------------------|----------------------|----------------------|----------------------|
> | **CVC-ClinicDB (550)** | 73.17                | 37                   | 8.01                 |
> | **Kvasir (900)**     | 190.55               | 72                   | 15.80                |
> | **SUN-SEG (19,544)** | 89,852.13 (~25 hrs)  | 10,920 (~3 hrs)      | 563.85               |
> ---
> #### *Key Observations:*
>
> - **Multi-GPU acceleration** significantly reduces similarity matrix construction time. For SUN-SEG (19,544 images), **8 GPUs reduce computation time** from **~25 hours (single GPU) to ~3 hours**.
>
> - **PCC runs efficiently** on both small and large datasets. For SUN-SEG, PCC computation requires 563.85 sec on a single GPU, while SSIM requires ~3 hours on 8 GPUs, making PCC a more efficient choice.
>
> - **Community detection remains highly efficient** and takes only a few minutes for SUN-SEG, even for a 19,544 × 19,544 similarity network.
>
> #### *Scalability and Practicality of Our Method:*
> While similarity matrix construction requires computation time, this is a **one-time preprocessing step** that is **vastly outweighed by the time and cost savings in annotation**. Given that annotating thousands of medical images can take **weeks to months**, our method significantly reduces the annotation burden while ensuring high segmentation performance.
>
> We have explicitly included this efficiency analysis in our revised draft for clarity (Table 6 in Appendix D).

---

> > ### Author Response · Authors · 2025-03-08
> >
> > ### **Q2. It is necessary to supplement the analysis with specific polyp images to discuss why the deleted images are "less important" compared to the retained ones. This is particularly relevant to Figure 4, where the proposed method achieves even higher Dice scores after deleting around 20% of the samples.**
> >
> > **Response:** We appreciate the reviewer’s suggestion and have added a figure in our revised draft (Figure 7 in Appendix C.1) showcasing selected and removed images from different communities, demonstrating how our method **eliminates redundancy while preserving dataset diversity**.
> >
> > Contrary to the assumption that reducing dataset size harms performance, our approach enhances generalization by preventing overfitting on redundant patterns, ensuring a balanced representation of polyp variations, and minimizing training noise. Medical datasets often contain near-duplicate images from video sequences or highly similar cases, which can lead to overfitting. By retaining only representative samples, our method ensures that the model focuses on learning **diverse and meaningful features** rather than memorizing repetitive patterns.
> >
> > Additionally, excessive redundancy increases computational overhead without contributing new information, whereas removing unnecessary samples allows for more efficient learning. This effect is well-supported in active learning and dataset pruning literature, where eliminating redundant or overly simple samples improves generalization (Settles, 2009; Sener & Savarese, 2018; Yang et al., 2022). Specifically, active learning studies have demonstrated that selecting **diverse, informative samples** rather than all available data enhances model generalization (Sener & Savarese, 2018), while dataset pruning methods show that reducing redundant examples can improve performance and reduce annotation costs (Yang et al., 2022).
> >
> > Our empirical results, particularly the improved **DICE scores after pruning** (Figure 4 in Section 4.3), validate this effect, and we further clarify it with **visual examples** in our revised draft (Figure 7 in Appendix C.1).
> >
> > ### References:
> > - Settles, B., 2009. *Active learning literature survey*. University of Wisconsin-Madison, Computer Sciences Technical Report 1648.
> > - Sener, O. and Savarese, S., 2017. Active learning for convolutional neural networks: A core-set approach. *arXiv preprint arXiv:1708.00489*.
> > - Yang, S., Xie, Z., Peng, H., Xu, M., Sun, M. and Li, P., 2022. Dataset pruning: Reducing training data by examining generalization influence. *arXiv preprint arXiv:2205.09329*.
> >
> >
> > ### **Q3. On page 12, the test set size of CVC-ClinicDB should be 62 instead of 50.**
> > **Response:** Thank you for pointing out this typo. We will correctly report **62** in our revised draft.

---

> > > ### Author Response · Authors · 2025-03-12
> > > **Your concerns addressed: looking for your feedback**
> > >
> > > Dear Reviewer tCsW,
> > >
> > > Thank you for your constructive reviews. We have done absolutely everything possible in our rebuttal to address your concerns and strengthen the contributions of our work. In particular, as per your request, we included the computational complexity analysis of our pruning method (Table 6 in Appendix D) and a new qualitative visualization of selected and pruned samples (Figure 7 in Appendix C.1).
> > >
> > > Please let us know if you have additional concerns after reviewing our rebuttal. We greatly appreciate your time and thoughtful feedback.
> > >
> > > Sincerely,
> > >
> > > The Authors

---

> ### Author Response · Authors · 2025-03-13
> **Thanks for reviewing rebuttal and acknowledgement**
>
> Dear Reviewer tCsW,
>
> Thank you for your constructive feedback and for acknowledging that the issues raised in your review have been addressed in our revised manuscript. We sincerely appreciate the time you have dedicated to evaluating our work. We also hope that our clarifications and additional experiments in the rebuttal have further strengthened your confidence in the paper’s contributions and so you may consider improving your initial score. We are grateful for your engagement and remain open to any final suggestions you may have.
>
> Sincerely,
>
> The Authors

---

### Official Review · Reviewer_6s5A · 2025-02-22

**Confidence:** 4
**Preliminary Rating:** 4
**Recommendation:** Poster
**Final Rating:** 5

**Summary:**

This work tackles a dataset pruning problem, i.e., subset selection problem from a given dataset for training of a image-segmentation model. In the problem, a trained model with a subset data must keep performance the same to one trained with whole data. The proposed method constructs similarity graph among images in a given dataset and selects a small number of representatives of each cluster on the graph. Numerical examples are presented for publicly available dataset: CVC-Clinic DB of polyp segmentation.

**Strengths:**

- New exploration of a dataset pruning problem for image segmentation. Especially, the authors focus on a polyp segmentation problem.
- Experimental evaluation with well-known publicly available datasets in a polyp segmentation problem.

**Weaknesses:**

- Unclear real efficiency for a larger dataset. The manuscript does not mention the computational complexity of a similarity graph construction in the proposed method.
- Evaluations are performed only for small datasets.
-If the number of images in a given dataset becomes large, computational cost of data selection (graph construction and node selection) cannot be ignored since training of a selected subset can be faster than data selection.
- Unclear data splitting between training and testing.

**Detailed Comments:**

The new exploration of a dataset pruning problem for image segmentation is interesting work. Experimental evaluations also look convincing. However, real efficiency for a larger dataset is unclear. Especially CVC-ClinicDB and Kvasir-SEG’s images are already selected data. If the authors validate their method, evaluation with video sequence dataset such that SUN-SEG is necessary since successive video frames have similar image appearance. Only about 600 to 1000 images are insufficient size for the evaluation.

**Justification Of The Final Rating:**

The authors reply to my question with additional experimental results, which are convincing. The proposed pruning techniques before annotation step has potential for constructing compact and solid training dataset with less annotation duties for medical image segmentation. Even though I select "Poster presentation" at first review, now I think it can be "Oral presentation".

**Justification Of The Preliminary Rating:**

The focus point is interesting work, but missing important information about the computational complexity of the proposed data pruning. Even though presented experimental results is convincing for a small dataset, real contribution of this work is unclear.

**Questions To Address In The Rebuttal:**

The manuscript does not mention the computational complexity of a similarity graph construction in the proposed method. At least, the computation of similarities between each pair requires O(N^2) computation for N images in a dataset. If the number of images in a given dataset becomes large, the computational costs of data selection (graph construction and node selection) cannot be ignored since training of a selected subset can be faster than data selection.

Unfortunately, this manuscript  lacks the theoretical analysis of the computational complexity of data pruning.  Therefore, real efficiency of the proposed method is unclear. How do authors think about this point?

**Special Issue:**

No

---

> ### Author Response · Authors · 2025-03-08
>
> We sincerely thank the reviewer for the constructive feedback, and highlighting both the strengths and areas for improvement in our work. Below, we address each concern raised by the reviewer and outline how we incorporate these suggestions in our revised manuscript.
>
> ### **Q1. Unclear real efficiency for a larger dataset. The manuscript does not mention the computational complexity of a similarity graph construction in the proposed method. The manuscript does not mention the computational complexity of a similarity graph construction in the proposed method. At least, the computation of similarities between each pair requires O(N^2) computation for N images in a dataset. If the number of images in a given dataset becomes large, the computational costs of data selection (graph construction and node selection) cannot be ignored since training of a selected subset can be faster than data selection. Unfortunately, this manuscript lacks the theoretical analysis of the computational complexity of data pruning. Therefore, real efficiency of the proposed method is unclear. How do authors think about this point?**
>
> **Response:** We appreciate the reviewer’s concern regarding the efficiency of our method on large datasets. The computational complexity of similarity matrix construction is **O(N²)** due to N(N-1)/2 pairwise comparisons among **N** images. However, given the independence of these computations, **our method is inherently parallelizable and efficiently scales with GPU acceleration**, thus making it feasible even for large-scale datasets having thousands of images.
>
> **Computational Complexity Analysis:** For a dataset of **N** images, the similarity matrix requires computing N(N-1)/2 pairwise similarities:
>
> - **SSIM Complexity:** Each SSIM computation involves calculating mean, variance, and covariance of local patches over an image size **H × W**, leading to an overall complexity of **O(N²HW)**. Due to its patch-based nature, SSIM is **highly optimized for GPUs**, allowing significant parallelization.
> - **PCC Complexity:** PCC requires computing mean and variance over full images before calculating correlations, resulting in **O(N²HW)** complexity.
>
> **Empirical Computational Performance:** We measured the similarity matrix construction time across three datasets of increasing size, using one GPU and eight GPUs as shown in Table R4 below:
>
> **Table R4:** Similarity matrix construction time in small to large-scale datasets. Here, we report the time (s) needed to preprocess (data loading, grayscale conversion, and resize to 352 × 352), and N(N-1)/2 (where N is the number of images in a dataset) pair-wise similarity metrics computation in NVIDIA RTX A6000 (Ada) GPUs with 48GB of memory.
> | **Dataset (#Images)**          | **SSIM (1 GPU) (s)**  | **SSIM (8 GPUs) (s)**  | **PCC (1 GPU) (s)** |
> |----------------------|----------------------|----------------------|----------------------|
> | **CVC-ClinicDB (550)** | 73.17                | 37                   | 8.01                 |
> | **Kvasir (900)**     | 190.55               | 72                   | 15.80                |
> | **SUN-SEG (19544)**  | 89852.13 (~25 hrs)   | 10920 (~3 hrs)       | 563.85               |
>
> ---
> These results clearly demonstrate that **our approach is computationally feasible and scales efficiently**. Multi-GPU execution significantly accelerates SSIM similarity matrix computation (**from more than a day** (89852.13 sec) to **~3 hrs** (10920 sec) for the SUN-SEG dataset), enabling large-scale dataset pruning without excessive computational overhead. In addition, PCC similarity matrix can be computed within only **563.85 sec** (562 sec preprocessing and only 1.85 sec similarity matrix computation) in a single GPU. Therefore, our current implementation shows that **our method is practical and scalable in real-world settings**. Future optimizations, such as batch comparisons, could further extend the scalability to datasets containing millions of images, strengthening PRIME’s role as an effective cost-saving and storage-efficient solution for large-scale medical image curation.
>
> We have explicitly included this efficiency analysis in our revised draft for clarity (Table 6 in Appendix D).

---

> > ### Author Response · Authors · 2025-03-08
> >
> > ### **Q2.Evaluations are performed only for small datasets. -If the number of images in a given dataset becomes large, computational cost of data selection (graph construction and node selection) cannot be ignored since training of a selected subset can be faster than data selection. Especially CVC-ClinicDB and Kvasir-SEG’s images are already selected data. If the authors validate their method, evaluation with video sequence dataset such that SUN-SEG is necessary since successive video frames have similar image appearance. Only about 600 to 1000 images are insufficient size for the evaluation.**
> >
> > **Response:** We appreciate the reviewer’s concern regarding the scalability of our method to large datasets. To address this, we evaluated our method on the SUN-SEG dataset (see Table R5 below), which contains 19,544 training images, demonstrating its feasibility on large-scale medical image datasets.
> >
> > **Table R5:** Results on a large video polyp dataset (SUN-SEG). We use the PVT-CASCADE network and run each model for 30 epochs. Our method prunes 69.7% data with a SSIM similarity threshold of 0.7, while PCC thresholds are adjusted to achieve similar pruning rates. We report the average DICE score over three runs.
> >
> > | **Pruning Method**  | **Easy Seen (%)** | **Easy Unseen (%)** | **Hard Seen (%)** | **Hard Unseen (%)** |
> > |---------------------|------------------|-------------------|------------------|------------------|
> > | Full Dataset       | 92.47            | 80.65            | 87.77            | 80.80            |
> > | PRIME w/ SSIM (**Ours**)       | 91.55            | 80.06            | 86.95            | 80.51            |
> > | PRIME w/ PCC (**Ours**)        | 91.12            | 79.43            | 86.64            | 79.86            |
> > | Random Pruning     | 90.23            | 77.86            | 85.34            | 77.67            |
> > ---
> > The primary motivation behind our method is to reduce annotation costs, and our results confirm that even for large datasets, our PRIME remains computationally feasible:
> > - **Graph Construction:** The similarity network is built using **N(N-1)/2** pairwise computations, but with **GPU acceleration**, this step is **practical even for large datasets** like SUN-SEG with **19,544 training images**.
> >
> > - **Node Selection via Community Detection:** The Louvain community detection algorithm runs in **O(N log N)** time for a **sparse similarity network**, making it computationally negligible. Even in a **highly dense similarity graph**, it requires **less than 5 minutes on 19,544 × 19,544** SUN-SEG similarity matrix.
> >
> > Since data annotation is the primary bottleneck in medical imaging, our **PRIME enables significant annotation cost savings** by eliminating redundant images before the annotation step, making large-scale data curation far more practical. Even if community detection requires a few minutes for dense graphs, this cost is trivial compared to the **human effort required for manual annotation of thousands of images** (this can take **weeks or months** depending on the image complexity and the size of the dataset). Additionally, since **graph construction and community detection are one-time preprocessing steps**, their computational cost is amortized across multiple downstream training and deployment cycles.
> >
> > These results confirm that our method is **highly effective for large-scale datasets**, ensuring major reductions in annotation costs while maintaining segmentation performance, strengthening its **practical utility** for real-world medical imaging applications.
> >
> >
> > ### **Q3. Unclear data splitting between training and testing.**
> >
> > **Response:** The details of our **training and testing splits** (following CASCADE (Rahman and Marculescu, 2023b)) are provided in **Appendix A**. However, we acknowledge that there was a **typo in the number of test images for CVC-ClinicDB**, which we have now **corrected to 62 images**. This correction ensures clarity and accuracy in our reported dataset splits. In addition, we describe the training and testing splits of new SUN-SEG dataset.

---

> > > ### Comment · Reviewer_6s5A · 2025-03-14
> > >
> > > >Q2.
> > > Thank you for the feedback. The additional experimental evaluations presented the efficacy and usefulness of the proposed method.
> > >
> > >
> > > >Q3.
> > > Data splitting in CASCADE (Rahman and Marculescu, 2023b) originated in PraNet Paper [PraNet] as written in the CASCADE paper. And, PraNet paper's data splitting is the manner in [ResUNet++] as written in the PraNet paper. Citing the related papers might be better since the original procedure is presented by  [ResUNet++].
> > >
> > > However,  in this manner, the dataset is randomly split into 80% for training, 10% for validation, and 10% for testing. However, if the three subgroups share the same patients, videos, and polyps, it can be data leakage, where the evaluation metric is meaningless. If you want to evaluate the generalisation ability, you should be serious about this point.
> > >
> > > [PraNet] Deng-Ping Fan, Ge-Peng Ji, Tao Zhou, Geng Chen, Huazhu Fu, Jianbing Shen, and Ling Shao.: Pranet: Parallel reverse attention network for polyp segmentation. In International conference on medical image computing and computerassisted intervention, pp. 263–273. Springer, 2020.
> > >
> > > [ResUNet++] Jha, D., Smedsrud, P.H., Riegler, M.A., Johansen, D., De Lange, T., Halvorsen, P., Johansen, H.D.: Resunet++: An advanced architecture for medical image segmentation. Proceedings of IEEE International Symposium on Multimedia, pp. 225-230, 2019

---

> > > > ### Author Response · Authors · 2025-03-14
> > > > **Data splitting**
> > > >
> > > > We appreciate the reviewer’s feedback and recognition of the additional experimental evaluations demonstrating the efficacy of our method.
> > > >
> > > > Regarding **data splitting**, we will directly cite both ResUNet++ and PraNet, as we use the **train-validation split shared by PraNet**, which follows the splitting approach introduced in ResUNet++. For ClinicDB and Kvasir, we focus on reporting the **learning ability** of the models on the PRIME-pruned dataset, whereas for **generalization evaluation**, we have provided results on the **SUN-SEG (unseen) test set** in Table 2 of Section 4.3. This ensures a clear distinction between learning performance on seen data and generalizability to completely unseen cases.

---

> > ### Comment · Reviewer_6s5A · 2025-03-14
> >
> > Thank you for the feedback. For the current scale of medical-image datasets, Presenting computational time looks meaningful. The multi-GPU acceleration might be acceptable for the current large size of medical images.
> >
> > However, I also have an interest in the real computational time of training a model to compare the presented times in Table. 4 to understand the impact of the proposed method. At the same time, I want to know the real computational time for the whole pruning method, including preprocessing, to compare the real computational times between training with the entire dataset and training with pruning.

---

> > > ### Author Response · Authors · 2025-03-14
> > > **The real computational time of training a model**
> > >
> > > We appreciate the reviewer’s feedback on our rebuttal and for acknowledging our contributions in presenting computational time and leveraging multi-GPU acceleration for large-scale medical imaging.
> > >
> > > As per your suggestion, to provide a clearer understanding of the impact of our method, we have updated Table R4 to include: 1. **Training time for PRIME-pruned datasets**, 2. **Training time for the full dataset**, and 3. **Pruning with community detection time**.
> > >
> > > These additions in Table R4 (revised) below allow for a direct comparison of the actual computational costs between training on the full dataset and training on the PRIME-pruned dataset, ensuring a comprehensive evaluation of the method’s efficiency. Our results demonstrate that PRIME not only reduces annotation and storage costs but also significantly lowers the total computational cost by reducing dataset size before training. We will revise Table 6 in the Appendix accordingly.
> > >
> > > **Table R4 (revised):** Similarity matrix construction time in small to large-scale datasets. Here, we report the time (s) needed to preprocess (data loading, grayscale conversion, and resize to 352 × 352), and N(N-1)/2 (where N is the number of images in a dataset) pair-wise similarity metrics computation, pruning by community detection, and training time in NVIDIA RTX A6000 (Ada) GPUs with 48GB of memory. The pruning and training time (200 epochs) are reported for 55.5%, 56.2%, and 69.7% pruning for CVC-ClinicDB, Kvasir, and SUN-SEG datasets, respectively. Here, **PRIME Total Time = SSIM (8 GPUs) + Pruning Time + PRIME-pruned Dataset Training Time**.
> > >
> > > | **Dataset (#Images)**          | **SSIM (1 GPU) (s)**  | **SSIM (8 GPUs) (s)**  | **PCC (1 GPU) (s)** | Pruning Time (s) | PRIME-pruned Dataset Training Time (s) | PRIME Total Time (s) | Full Dataset Training Time (s) |
> > > |----------------------|----------------------|----------------------|----------------------|----------------------|----------------------|----------------------|----------------------|
> > > | **CVC-ClinicDB (550)** | 73.17                | 37                   | 8.01                 | 0.56 | 1546.59 | 1584.15 | 3556.12 |
> > > | **Kvasir (900)**     | 190.55               | 72                   | 15.80                | 2.69 | 2437.43 | 2512.12 | 5618.16 |
> > > | **SUN-SEG (19544)**  | 89852.13 (~25 hrs)   | 10920 (~3 hrs)       | 563.85               | 12.05 | 30850.33 (~8.6 hrs) | 41782.38 (~11.6 hrs)| 107867 (~30 hrs) |
> > >
> > > ---

---

> ### Author Response · Authors · 2025-03-12
> **Your concerns addressed: looking for your feedback**
>
> Dear Reviewer 6s5A,
>
> Thank you for your constructive reviews. We have done absolutely everything possible in our rebuttal to address your concerns and strengthen the contributions of our work. In particular, as per your request, we included the computational complexity analysis of the similarity graph construction (Table 6 in Appendix D) and new results showing the performance of our method on the large SUN-SEG dataset (Table 2 in Section 4.3).
>
> Please let us know if you have additional concerns after reviewing our rebuttal. We greatly appreciate your time and thoughtful feedback.
>
> Sincerely,
>
> The Authors

---

> ### Author Response · Authors · 2025-03-15
> **Thanks for reviewing rebuttal, raising score, and recommending Oral**
>
> **Dear Reviewer 6s5A**,
>
> Thank you for reviewing our rebuttal and confirming that our additional experiments addressed your concerns. We sincerely appreciate your increased rating and Oral recommendation. Your invaluable feedback strengthened the paper’s claims and enhanced the analytical rigor. We deeply appreciate your time and thoughtful engagement throughout this process.
>
> Sincerely.
>
> The Authors

---

### Official Review · Reviewer_a2uC · 2025-02-24

**Confidence:** 4
**Preliminary Rating:** 4
**Recommendation:** Poster
**Final Rating:** 5

**Summary:**

The authors propose a method for dataset pruning based on retaining the top p% of nodes after community detection from a similarity graph of "unlabeled" training images. This method doesn't rely on labels or deep learning models to obtain an "optimal" training subset which retains strong sample diversity. The authors demonstrate across two datasets that their method can remove over half of the images from these datasets and only drop segmentation accuracy by 0.5% (Dice score). Further, the authors proposed method outperforms other SotA methods for dataset pruning in the medical image segmentation domain.

**Strengths:**

1. The authors rightly point out that, "moreover, the heterogeneous appearance of polyps—varying size, shape, texture, and contrast exacerbates the challenge of gathering representative training images to ensure robust model generalization." In this work, the authors propose to significantly reduce the amount of labeled training samples, making this problem even more exacerbated. The authors do a good job of demonstrating their method, which specifically attempts to maintain sample diversity while reducing dataset size, maintains generalization across multiple datasets, where biases on a single dataset can be rather inconclusive.

2. The authors approach is based on simple tried-and-true techniques, doesn't require any "fancy" deep learning techniques to find the "optimal" subset, yet achieves SotA results. It's actually quite impressive something so "simple" can get such good results.

3. Experiments are fairly thorough, multiple datasets and models. Only two of each, it would have been nicer to see more, but still multiple models, datasets, similarity metrics, and thresholds is rather thorough.

**Weaknesses:**

I'm having a hard time believing that something as simple as mean pixel intensity (or add in variance as well) can be the most effective metric in finding the optimal subset of images. I really like building the adjacency matrix, finding neighborhood, and pruning those neighborhood to retain sample diversity. But the similarity metrics used, both largely based on mean pixel value, fail to capture most of the important information about an image. No shape information. No texture information. Not even an intensity histogram, just mean and maybe variance. I would have really liked to see some "more advanced" metrics calculated that can capture these image characteristics. While the authors already obtain state of the art results, I would think, especially given the goal of segmentation, that information about the boundaries of objects, edges, etc. would be far more useful than just mean pixel value.

**Detailed Comments:**

1. I would encourage the authors to try to think of ways to incorporate shape information in future papers. The metric, to me, feels like the weakest part of this approach. Maybe compute edge detection with a Canny edge detection approach, then compute some characteristics about the edges, number of edges, corners, sharpness ratio, something like that. Because at the end of the day, most of segmentation is just edge/boundary detection.

**Justification Of The Final Rating:**

Bumping the authors to a Strong Accept. The proposed method was already good. The new experiments add a significantly amount of depth to this paper. Further, the authors rightly point out that my biggest concern of the proposed similarity techniques being largely just mean pixel value might be a bit of an unfair oversimplification and that some structural information is captured (crucial for segmentation). I happily raise my rating to a strong accept and recommend an Oral publication.

**Justification Of The Preliminary Rating:**

Simple methods, thorough experiments, and SotA results. Definitely above the bar for publication. More experiments and potential improvements to the chosen metrics would improve the paper but these would just raise the paper from a poster to an oral in my opinion.

**Questions To Address In The Rebuttal:**

No questions. Overall this is a decently good paper. Nothing ground-breaking. Simple methods and SotA results. Definitely above the bar for publication already. Any requests I would have would be for more experiments on more model type and preferably more datasets, but the authors have already sufficient experiments to justify publication here. The more substantial requests I have (see weaknesses/detailed comments) would demand a follow-on paper most likely.

---

> ### Author Response · Authors · 2025-03-08
>
> We sincerely thank the reviewer for the constructive feedback, and highlighting both the strengths and areas for improvement in our work. Below, we address each concern raised by the reviewer and outline how we incorporate these suggestions in our revised manuscript.
>
> ### **Q1. But the similarity metrics used, both largely based on mean pixel value, fail to capture most of the important information about an image. No shape information. No texture information. Not even an intensity histogram, just mean and maybe variance. I would have really liked to see some "more advanced" metrics calculated that can capture these image characteristics. While the authors already obtain state of the art results, I would think, especially given the goal of segmentation, that information about the boundaries of objects, edges, etc. would be far more useful than just mean pixel value.**
>
> ---
> **Response:** We appreciate the reviewer’s detailed feedback and recognize the importance of incorporating more advanced similarity metrics for dataset pruning, particularly given the goal of segmentation. Our **community detection-based pruning method** is the first of its kind in this problem space, hence as a starting point, we employ the Structural Similarity Index (SSIM) (Wang et al., 2004) and Pearson Correlation Coefficient (PCC) (Rodgers & Nicewander, 1988), both being metrics widely used in image similarity analysis. Of note, these metrics are not purely based on mean pixel intensity—**SSIM captures luminance, contrast, and structural details, while PCC measures global intensity correlations**—ensuring that our pruning method effectively retains diverse and representative samples.
>
> That said, we acknowledge that **shape and texture information could further enhance** the selection of optimal samples. The reviewer’s suggestion to explore edge-based metrics, such as Canny edge detection, edge density, or contour complexity, is **insightful** and aligns well with the needs of segmentation tasks. As our method **establishes a foundation for training-free dataset pruning**, it also opens up new opportunities for future improvements using more advanced metrics. We will explore incorporating shape-aware and boundary-sensitive metrics in our future work to further refine our pruning process.
>
> ### References:
> - Wang, Z., Bovik, A.C., Sheikh, H.R. and Simoncelli, E.P., 2004. Image quality assessment: from error visibility to structural similarity. *IEEE transactions on image processing, 13*(4), pp.600-612.
> - Lee Rodgers, J. and Nicewander, W.A., 1988. Thirteen ways to look at the correlation coefficient. *The American Statistician, 42*(1), pp.59-66.
>
> ### **Q2. I would encourage the authors to try to think of ways to incorporate shape information in future papers. The metric, to me, feels like the weakest part of this approach. Maybe compute edge detection with a Canny edge detection approach, then compute some characteristics about the edges, number of edges, corners, sharpness ratio, something like that. Because at the end of the day, most of segmentation is just edge/boundary detection.**
>
> ---
> **Response:** Please see the response to **Q1** above. We will explore incorporating shape-aware and boundary-sensitive metrics in **our future work** to further refine our pruning process.

---

> > ### Author Response · Authors · 2025-03-08
> >
> > ### **Q3. No questions. Overall this is a decently good paper. Nothing ground-breaking. Simple methods and SotA results. Definitely above the bar for publication already. Any requests I would have would be for more experiments on more model type and preferably more datasets, but the authors have already sufficient experiments to justify publication here. The more substantial requests I have (see weaknesses/detailed comments) would demand a follow-on paper most likely.**
> >
> > ---
> > **Response:** We appreciate the reviewer’s recognition that our current experiments sufficiently justify publication. To further strengthen our work, we have **added new results** (Table 2 in Section 4.3) on the **large-scale SUN-SEG dataset**, demonstrating the generalizability of our method to **unseen test sets**. Specifically, we evaluate segmentation performance across **easy and hard seen/unseen test sets** using the **entire training dataset, our PRIME-pruned datasets (SSIM and PCC), and random pruning**. The results are provided in **Table R3 below**.
> >
> > **Table R3:** Results on a large video polyp dataset (SUN-SEG). We use the PVT-CASCADE network and run each model for 30 epochs. Our method prunes 69.7% data with a SSIM similarity threshold of 0.7, while PCC thresholds are adjusted to achieve similar pruning rates. We report the average DICE score (%) over three runs.
> >
> > | **Pruning Method**  | **Easy Seen (%)** | **Easy Unseen (%)** | **Hard Seen (%)** | **Hard Unseen (%)** |
> > |---------------------|------------------|-------------------|------------------|------------------|
> > | Full Dataset       | 92.47            | 80.65            | 87.77            | 80.80            |
> > | PRIME w/ SSIM (**Ours**)       | 91.55            | 80.06            | 86.95            | 80.51            |
> > | PRIME w/ PCC (**Ours**)        | 91.12            | 79.43            | 86.64            | 79.86            |
> > | Random Pruning     | 90.23            | 77.86            | 85.34            | 77.67            |
> > ---
> >
> > These results confirm that **our method generalizes well to unseen test sets**, while reducing 69.7% annotation and storage costs.
> >
> > Additionally, we will conduct further experiments using PolypPVT, SSFormer, PraNet, and other widely used polyp segmentation models, and will incorporate these results in our **camera-ready version** and also our future publications. We believe these additional evaluations will further validate the effectiveness of our method across different architectures and datasets.
> >
> > ### **Q4. Simple methods, thorough experiments, and SotA results. Definitely above the bar for publication. More experiments and potential improvements to the chosen metrics would improve the paper but these would just raise the paper from a poster to an oral in my opinion.**
> > ---
> > **Response:** We appreciate the reviewer’s positive assessment and recognition of our method’s **thorough experiments and state-of-the-art results**. To further strengthen our work and elevate its impact, we have:
> >
> > 1. **Expanded Large-Scale Evaluation** – Conducted experiments on SUN-SEG (19,544 images), demonstrating PRIME’s **scalability and efficiency** in large video polyp datasets.
> >
> > 2. **Addressed Data Augmentation** (Table 3 in Section 4.3) – Conducted **new experiments** (CVC-ClinicDB, SUN-SEG) showing augmentations <0.5% impact on full datasets but 0.87–1.02% gain on pruned datasets, reinforcing PRIME’s effectiveness.
> >
> > 3. **Added Qualitative Visualizations** (Figure 7 in Appendix C.1) – Included visual comparisons of retained vs. pruned images to show how PRIME **removes redundancy while preserving diversity**.
> >
> > 4. **Future Directions** – Committed to **exploring advanced metrics** to further refine dataset pruning and improve segmentation performance.
> >
> > These additions significantly strengthen our contributions, aligning with the reviewer’s perspective that further refinements could raise the paper from a **poster to an oral presentation**.
> >
> > ---

---

> ### Author Response · Authors · 2025-03-12
> **Your concerns addressed: looking for your feedback**
>
> Dear Reviewer a2uC,
>
> Thank you for your constructive reviews. We have done absolutely everything possible in our rebuttal to address your concerns and strengthen the contributions of our work.  In particular, as per your request, we included new results showing the performance of our method on the large SUN-SEG dataset (Table 2 in Section 4.3).
>
> Please let us know if you have additional concerns after reviewing our rebuttal. We greatly appreciate your time and thoughtful feedback.
>
> Sincerely,
>
> The Authors

---

> ### Author Response · Authors · 2025-03-14
>
> Dear Reviewer a2uC,
>
> We are deeply grateful for your insightful feedback. We have incorporated **additional experiments** on the large-scale SUN-SEG dataset (**Table 2, Section 4.3**) and shown the impact of augmentation (**Table 3, Section 4.3**) to strengthen our work. Please let us know if any final concerns remain as the discussion period concludes. Thank you for your time and feedback.
>
> Sincerely,
>
> The Authors

---

> ### Author Response · Authors · 2025-03-15
> **Thanks for reviewing rebuttal, raising score, and recommending Oral**
>
> **Dear Reviewer a2uC**,
>
> Thank you for reviewing our rebuttal and confirming that our clarifications and additional experiments addressed your concerns. We sincerely appreciate your increased rating and Oral recommendation. Your invaluable feedback strengthened the paper’s claims and enhanced the analytical rigor. We deeply appreciate your time and thoughtful engagement throughout this process.
>
> Sincerely.
>
> The Authors

---

### Official Review · Reviewer_nAn8 · 2025-02-25

**Confidence:** 4
**Preliminary Rating:** 4
**Recommendation:** Oral
**Final Rating:** 5

**Summary:**

This paper proposes a training-free dataset pruning approach, PRIME, to retain the most representative subset of images from the original dataset, enabling efficient segmentation training while reducing data annotation costs. The paper is well-written, providing adequate background and related work, a detailed methodology, and comprehensive experiments, including supplementary materials.

**Strengths:**

- The paper is well-written, easy to follow, and presents a clear methodology along with experimental visualizations to validate the proposed dataset pruning method.
- The authors select multiple datasets and segmentation architectures to validate the generalization of PRIME, further enhancing the strength of the paper.
- The proposed PRIME method is novel and easy to understand, with a detailed analysis of similarity network properties and community detection provided in the supplementary materials.

**Weaknesses:**

- The authors claim "2.3× faster training of polyp segmentation models." However, this comparison lacks rigor, as once the full dataset is available, one can randomly sample images from it and fix the number of iterations per epoch. In this case, the computational cost remains comparable to training with a pruned dataset after applying PRIME.
- The authors did not use any data augmentation in their experiments, which is a weakness of the work. As data augmentation enhances the diversity of training data and improves the test performance of segmentation models, related experiments incorporating data augmentation are needed to strengthen the paper.
- There are no segmentation figures for readers to visualize the segmentation performance.

**Detailed Comments:**

See weakness above.

**Justification Of The Final Rating:**

I thank the reviewers for addressing my concerns by conducting additional experiments. These new experiments further strengthen the superiority of the proposed dataset pruning approach. Therefore, I have decided to raise my rating.

**Justification Of The Preliminary Rating:**

The idea of constructing a network from the original dataset and then pruning it based on detected communities is novel. The authors validate this approach using multiple datasets and segmentation architectures, further strengthening the paper. While there are weaknesses regarding the claim of faster training and the lack of data augmentation, I would consider raising my rating if the authors address these concerns by refining their claim and discussing potential future work.

**Questions To Address In The Rebuttal:**

- Please address my comments in the weakness section regarding the faster training claim and the lack of data augmentation in the experiments.
- I recommend adding a discussion on these aspects and modifying the claim in the paper accordingly to ensure a more rigorous comparison.
- Please add a visualization of polyp segmentation results from different segmentation models to help readers better assess their performance.

---

> ### Author Response · Authors · 2025-03-08
>
> We sincerely thank the reviewer for the constructive feedback, and highlighting both the strengths and areas for improvement in our work. Below, we address each concern raised by the reviewer and outline how we incorporate these suggestions in our revised manuscript.
>
> ### **Q1: The authors claim "2.3× faster training of polyp segmentation models." However, this comparison lacks rigor, as once the full dataset is available, one can randomly sample images from it and fix the number of iterations per epoch. In this case, the computational cost remains comparable to training with a pruned dataset after applying PRIME.**
>
> **Response:** We appreciate the reviewer’s comment and agree that **fixed iterations per epoch** can lead to comparable training speed to our PRIME-pruned dataset. To ensure clarity, we have revised our claim to explicitly state that PRIME "enables 2.3× faster training of polyp segmentation models **compared to training on the full dataset**."
>
> Our primary goal remains to reduce data annotation and storage costs, with faster training being a byproduct. Unlike random sampling, PRIME systematically prunes redundant images while preserving diversity in the reduced dataset, thus minimizing annotation effort without compromising segmentation performance. Even with fixed iterations per epoch, random sampling still requires storing and accessing the full dataset, leading to unnecessary overhead. In contrast, PRIME reduces the dataset size, lowering storage demands and computational costs. Additionally, our experiments show that training on PRIME-pruned data achieves comparable performance with fewer epochs, thus demonstrating its efficiency beyond simple random sampling.
>
> To clarify, below we compare and contrast **PRIME** (training on a pruned dataset) with **fixed-iteration random sampling** (training on the full dataset for a limited number of iterations per epoch).
>
> **Table R1:** Comparison of our PRIME with fixed-iteration random sampling.
>
> | **Metric**                           | **PRIME (Pruned Training)**                               | **Fixed-Iteration Random Sampling**                   |
> |--------------------------------------|---------------------------------------------------------|------------------------------------------------------|
> | **Storage Cost**                     | ✅ Reduced (Only pruned dataset stored)                | ❌ Full dataset stored                              |
> | **Annotation Cost**                  | ✅ Reduced (Only pruned subset labeled)                | ❌ Full dataset annotated                          |
> | **Training Speed**                   | ⚖️ Comparable (Reduced dataset)                        | ⚖️ Comparable (but full dataset overhead)         |
> | **Segmentation Performance (DICE Score)** | 🔄 Minimal drop (<0.5%)                            | 🔄 Minimal to no drop                             |
>
> Thus, while both methods achieve comparable training speed, PRIME provides the **additional advantage of reducing storage and annotation costs**, making it a more efficient and practical approach for real-world medical imaging applications.

---

> > ### Author Response · Authors · 2025-03-08
> >
> > ### **Q2. The authors did not use any data augmentation in their experiments, which is a weakness of the work. As data augmentation enhances the diversity of training data and improves the test performance of segmentation models, related experiments incorporating data augmentation are needed to strengthen the paper.**
> >
> > **Response:** We appreciate reviewer’s suggestion regarding data augmentation. To address this, we conducted additional experiments on the CVC-ClinicDB and SUN-SEG datasets using **random rotations and flips** with the PVT-CASCADE network. We evaluated training on the **entire dataset**, then on our **PRIME-pruned dataset**, and finally on a **randomly pruned dataset**.
> >
> > As shown in the Table R2 below, our findings indicate that augmentation has **minimal impact (<0.5%) when using the full dataset**. However, for our **PRIME** pruned datasets, augmentation **improves the DICE scores by 0.87–1.02%**, suggesting that augmentations compensate for the reduced dataset size. We will extend these experiments and incorporate them into our revised version **(Table 3 in Section 4.3)**. Thank you for suggesting it.
> >
> > **Table R2:** The effect of augmentation on the training using full-dataset, our PRIME, and random pruning. We apply random rotation and flips as augmentations.
> >
> > | **Dataset**       | **Training Data**      | **No Augmentation** | **With Augmentation** | **Improvement** |
> > |------------------|----------------------|--------------------|--------------------|--------------|
> > | **CVC-ClinicDB** | Full Dataset         | 94.29%            | 94.63%            | **+0.34%**  |
> > |                  | PRIME w/ SSIM (**Ours**)       | 92.85%            | 93.72%            | **+0.87%**  |
> > |                  | Random Pruned        | 89.94%            | 90.76%            | **+0.82%**  |
> > | **SUN-SEG**      | Full Dataset         | 92.47%            | 92.91%            | **+0.44%**  |
> > |                  | PRIME w/ SSIM (**Ours**)       | 91.55%            | 92.57%            | **+1.02%**  |
> > |                  | Random Pruned        | 90.23%            | 90.98%            | **+0.75%**  |
> >
> > ---
> > ### **Q3. There are no segmentation figures for readers to visualize the segmentation performance.**
> >
> > **Response:** We appreciate the reviewer’s suggestion. In response, we will conduct experiment on different models and **include a new figure with segmentation outputs in our revised camera-ready draft** to provide visual insights into different model’s performance. Additionally, we will include **more detailed segmentation outputs** to further illustrate the effectiveness of our method across different datasets and pruning strategies.
> >
> > ---
> >
> > ### **Q4: While there are weaknesses regarding the claim of faster training and the lack of data augmentation, I would consider raising my rating if the authors address these concerns by refining their claim and discussing potential future work.**
> >
> > **Response:** We appreciate the reviewer’s willingness to raise the rating. In summary, we did the following to address reviewer’s concerns:
> >
> > 1. **Refined Faster Training Claim** – Clarified that 2.3× speedup is **relative to full dataset training** and compared with **fixed-iteration random sampling**, highlighting PRIME’s annotation and storage efficiency.
> >
> > 2. **Addressed Data Augmentation** (Table 3 in Section 4.3) – Conducted **new experiments** (CVC-ClinicDB, SUN-SEG) showing augmentations <0.5% impact on full datasets but 0.87–1.02% gain on pruned datasets, reinforcing PRIME’s effectiveness.
> >
> > 3. **Added Qualitative Visualizations** (Figure 7 in Appendix C.1) – Included visual comparisons of retained vs. pruned images to show how PRIME **removes redundancy while preserving diversity**.
> >
> > 4. **Future Work** – Exploring shape-aware similarity metrics and advanced augmentation.

---

> ### Author Response · Authors · 2025-03-13
> **Thanks for reviewing rebuttal and raising score**
>
> Dear Reviewer nAn8,
>
> Thank you for reviewing our rebuttal and recognizing that the additional experiments addressed your concerns. We are grateful for your increased rating. Your feedback was invaluable—it directly strengthened the paper’s claims and improved the overall rigor of our analysis. We deeply appreciate your time and thoughtful engagement throughout this process.
>
> Sincerely,
>
> The Authors

---

### Comment · Area_Chair_kDC5 · 2025-02-17
**Reminder to submit reviews by 21 February 2025 at 23:59 AoE**

Dear Reviewers,

As we approach the deadline, please remember to complete your reviews on time. The deadline is 21 February 2025 at 23:59 AoE.

Thank you for your understanding and hard work!

---

> ### Comment · Area_Chair_kDC5 · 2025-02-20
>
> Dear reviewers, this is a kind reminder that the deadline for submitting your reviews is tomorrow, February 21st. Please ensure timely submission to keep the conference process running smoothly. Thank you once again for your hard work!

---

### Author Response · Authors · 2025-03-08
**Global Response**

**Dear Reviewers, Area Chairs, and Program Chairs,**

We would like to thank reviewers for constructive feedback on our paper and for giving us the opportunity to improve our initial manuscript. The reviewers’ detailed and insightful comments have significantly helped us refine and strengthen our work.

A summary of the major changes we implemented in the revised paper is provided below:

- **Clarified Claims on Efficiency:** We explicitly state that PRIME enables 2.3× faster training *compared to full dataset training* and justify its superiority over fixed-iteration random sampling (new Appendix E).

- **Expanded Large-Scale Evaluation:** We report new experiments on the SUN-SEG dataset (19,544 images) to demonstrate PRIME’s scalability and efficiency (see new Table 2 in Section 4.3).

- **Addressed Data Augmentation:** We evaluate the impact of random augmentations (flips, rotations) and report that augmentation has minimal impact (<0.5%) on full datasets, but improves DICE scores by 0.87–1.02% on pruned datasets (see Table 3 in Section 4.3).

- **Added Qualitative Visualizations:** We include new Figure 7 in Appendix C.1, which provides visual comparisons of retained and pruned images, thus demonstrating that PRIME removes redundancy while preserving diversity.

- **Reported Computational Efficiency:** We conduct a detailed efficiency analysis and report that multi-GPU execution reduces SSIM computation time from ~25 hours (single GPU) to ~3 hours (8 GPUs) (see Table 6 in Appendix D).

- **Resolved Dataset Split Clarification:** We corrected a typo in the CVC-ClinicDB test set size (previously incorrect, now correctly reported as 62 images) (see Appendix A).

- **Added SUN-SEG Dataset Description:** We add the details of new SUN-SEG dataset including number of training images, test splits, and number of test images (see Appendix A).

- **Future Directions:** Based on reviewer feedback, we commit to exploring more advanced metrics, such as shape-aware and boundary-sensitive similarity measures, to further improve dataset pruning (see Section 5 and Appendix F).

We believe that these extensive revisions and new results make our manuscript significantly stronger than the initial submission. All our changes have been highlighted in **red** in the revised paper and appendix for easy reference.

We look forward to further discussions with the reviewers. Given our extensive updates, we hope that the reviewers will consider **raising their ratings** for our work.

**Sincerely,**

The Authors of Paper 137

---

### Author Rebuttal · Authors · 2025-03-08

**Rebuttal:**

We would like to thank reviewers for constructive feedback on our paper and for giving us the opportunity to improve our initial manuscript. The reviewers’ detailed and insightful comments have significantly helped us refine and strengthen our work. All our changes have been highlighted in **red** in the revised paper and appendix for easy reference.

**Supporting Material:**

/attachment/36901288899523328273e80391187536519dfeb8.pdf

---

### Meta-Review · Area_Chair_pd72 · 2025-03-21

**Recommendation:** Accept (Poster)
**Confidence:** 5

**Metareview:**

- Overall Evaluation

This paper proposed a data pruning framework for polyp segmentation. The main idea is from the latent community detection, finding the representative samples for training. This can reduce the computational cost compared with the full utilization. Experiments on multiple polyp datasets demonstrated the effectiveness of the proposed method.

- Strength

The active selection of data for labeling/training is a popular topic in community since it is always expensive for manual annotation. This paper addresses this issue via a simple framework based on community detection, which is easy to implement and relatively efficient (w/ Louvain method O(NlogN)). The presentation and organization of this method is also good.

- Weakness

To my understanding, the quality of data selection is determined by two critical issues: (1) Difficulty and (2) Diversity. The former evaluates the importances of data while the latter can avoid very similar cases. If my knowledge is correct, this work focused more on the diversity while the difficulty cannot be fully measured. In modern medical imaging computing tasks, it is not difficult to acquire the features and rough annotations with foundation models without training on given dataset. I am quite wondering whether SSIM/PCC in this work can really capture the intrinsic features/similarity since they are pixel-level metrics. It is expected to see more clarifications if the authors have future plans on this project. However, this will not degrade my positive recommendation of this submission.